# New Rabies Vaccines for Use in Humans

**DOI:** 10.3390/vaccines7020054

**Published:** 2019-06-20

**Authors:** Hildegund C. J. Ertl

**Affiliations:** The Wistar Institute, Philadelphia, PA 19104, USA; ertl@wistar.org; Tel.: +1-215-898-3863

**Keywords:** rabies virus, vaccines, post-exposure prophylaxis (PEP), pre-exposure prophylaxis (PrEP)

## Abstract

Although vaccines are available, rabies still claims more than 55,000 human lives each year. In most cases, rabies vaccines are given to humans after their exposure to a rabid animal; pre-exposure vaccination is largely reserved for humans at high risk for contacts with the virus. Most cases of human rabies are transmitted by dogs. Dog rabies control by mass canine vaccination campaigns combined with intensive surveillance programs has led to a decline of human rabies in many countries but has been unsuccessful in others. Animal vaccination programs are also not suited to control human rabies caused by bat transmission, which is common in some Central American countries. Alternatively, or in addition, more widespread pre-exposure vaccination, especially in highly endemic remote areas, could be implemented. With the multiple dose regimens of current vaccines, pre-exposure vaccination is not cost effective for most countries and this warrants the development of new rabies vaccines, which are as safe as current vaccines, but achieve protective immunity after a single dose, and most importantly, are less costly. This chapter discusses novel rabies vaccines that are in late stage pre-clinical testing or have undergone clinical testing and their potential for replacing current vaccines.

## 1. Introduction

Rabies continues to claim upwards of 55,00 human lives each year [1]. Most deaths occur in less developed countries in Asia and Africa, and disproportionally affect children below the age of 15 [1]. In most cases, the virus is transmitted through bites or licks by an infected dog. Unless the wound is promptly cleaned, and post-exposure prophylaxis (PEP) is administered, the human victim may develop an encephalitis that is nearly always fatal. Vaccines to rabies virus are available. They are based on fowl embryo- or tissue culture-grown inactivated virus. Rabies vaccines can be given preventatively to humans at high risk for exposure to the virus. In most cases the vaccines are given after exposure to a suspected rabid animal, and depending on the severity of the exposure, PEP must be combined with a rabies immunoglobulin (RIG) preparation of human or equine origin that is infiltrated into the wound. The vaccine is safe and efficacious but underused especially in developing countries. Socioeconomic factors lead to lack of appropriate vaccination of rabies-exposed humans. Rabies vaccines are costly and have to be given several times, which becomes very burdensome for those living in remote areas. RIG is even more expensive and in short supply. 

How can we reduce the death toll of human rabies? One option is to decrease rabies in the species that most commonly transmits the virus. Dogs are responsible for over 99% of human rabies cases. Mandatory dog rabies vaccination has virtually eliminated human rabies in Europe and most of the Americas [2]. Stray dogs are common in Asia and Africa and their vaccination requires either baiting by oral vaccines or parental vaccination upon capture of free-roaming dogs, which is time- and labor-intensive [3,4]. Alternatively, new human vaccines could be developed that achieve protective immunity after a single immunization, reduce the need for RIG if given after severe exposure and/or are cost effective if used for pre-exposure prophylaxis (PrEP) in highly endemic areas. Routine rabies PrEP would be especially useful for children living in remote areas with limited access to health care. This was demonstrated in Peru, a country that upon experiencing several human rabies outbreaks caused by vampire bats in Amazonia [5,6], implemented PrEP and thereby stopped further human deaths due to rabies [7].

## 2. Vaccine-induced Correlates of Protection 

Numerous studies have shown that protection against rabies virus infection is mediated by virus neutralizing antibodies (VNAs) against the viral glycoprotein [8,9,10] that is expressed as trimers on the surface of the virion. As a rule, VNA titers of 0.5 international units (IU) are considered to be protective [10], but this value should be viewed with caution. Lyssaviruses, which can all cause human rabies, are divided into 3 phylogroups, which are further divided into several genotypes [11]. Rabies vaccines are based on rabies lyssavirus, a phylogroup I, genotype 1 virus. The vaccine offers outstanding protection against rabies lyssavirus, which is the most common cause of human rabies. It protects against other genotypes of phylogroup I although VNA titers, which are needed for reliable protection against these viruses that are genetically less related to the vaccine than rabies lyssavirus, have not yet been established. Current vaccines fail to protect against lyssaviruses of phylogroups II and III. Human infections with these viruses, which are transmitted by bats, are exceedingly rare. It is thus unlikely that current vaccine will be modified to increase breadth of protection to all lyssaviruses [12]. 

## 3. Current Rabies Vaccines

Licensed rabies vaccines for human use are based on inactivated purified rabies virus grown either in tissue culture or in embryonated duck or chicken eggs. Nerve-tissue grown vaccines, which are less costly but are also less immunogenic and, even more importantly, can have serious side-effects, are no longer recommended by the World Health Organization (WHO) and their use has largely been discontinued. Rabies vaccines can be given intramuscularly (IM) or at a 5–10-fold lower dose intradermally (ID). A number of different regimens are approved for use in humans. For PrEP the vaccine is given typically three times on days 0, 7, and 21 or 28. Efforts are underway to change this to a two time point 2-dose regimen for which individuals are vaccinated into two sites on days 0 and 7. After exposure to a rabid animal, previously vaccinated individuals need to receive a boost; they do not require treatment with RIG. 

Unvaccinated individuals exposed to a rabid animal should receive, as soon as possible, thorough wound cleaning. Depending on the severity of exposure the wound should be infiltrated with RIG used at 20 mg/kg for human serum and 40 mg/kg for equine serum. Left-over serum that cannot be infiltrated into the wound due to space limitations should not be injected into a distant site. Individuals should then start vaccination. The vaccine can be given IM into one site on days 0, 3, and 7 followed by a fourth dose anytime between days 14–28. As an alternative IM regimen, the vaccine can be given to two sites on days 0 and 3 followed by one site injections on days 7 and 21. ID immunization should be given into two sites on days 0, 3, and 7 [13].

As already pointed out, egg- or tissue culture-grown rabies vaccines are safe, and if given correctly, highly effective in preventing disease and death. Nevertheless, they are costly. In the US PEP costs upwards of $3,000 and can cost as much as $40,000. It is far less expensive in developing countries where the vaccine, if given ID, costs about $10–15 while equine RIG for a 60 kg human would add another $20. Further costs through miscellaneous medical supplies, fees for health care providers, travel, and loss of wages further increases the overall expense, which in developing countries commonly exceeds the weekly or even monthly income of a family. The outcome, avoidance of PEP upon exposure to a rabid animal, is predictable and fatal.

## 4. Features that Would Improve Rabies Vaccines

One could envision the development of separate vaccines for rabies PEP and PrEP as they have different goals. A vaccine for PEP needs to induce VNAs as fast as possible to prevent rabies virus from spreading into the central nervous system. A PrEP vaccine on the other hand should induce sustained VNA titers and robust memory B and CD4^+^ T helper cell responses that allow for rapid VNA recall after a boost. Any type of new rabies vaccine, whether it is used for PEP, PrEP, or both, would have to be as safe and efficacious as our current vaccines. As rabies is fatal in more than 99.9% of humans that develop the disease, 80–90% vaccine efficacy, which is viewed as adequate for many of our current vaccines, such as influenza vaccines, which in some years are effective in less than 50% of recipients [14], is not acceptable for a rabies vaccine. Many vaccines work well in some people but not in others. Some vaccines show striking regional differences, for example the high efficacy rotavirus vaccines show in the US and Europe, is not being recapitulated in Africa [15]. The variability in human immune responses to vaccines is not fully understood and it is assumed that both genetic [16] and environmental factors, including the microbiome [17] and concurrent infectious disease burden [18], play a role. Needless to say, immune responses to a new rabies vaccine would have to be consistent throughout different human populations. 

A new rabies vaccine would have to cost less than currently licensed vaccines. Low cost is especially crucial for a PrEP rabies vaccine, which would need to cost no more than $1–3 in order to be cost-neutral to the alternative of treating exposed individuals with PEP. This takes into account that even after PrEP, a boost should be administered after exposure to a rabid animal. It does not take into account that cost-neutrality is influenced by the incidence of exposure to potentially rabid animals, which varies from country to country and even from region to region. For example, Kenya reported from 2002 to 2012 336 dog bites/100,000 persons [19], in 2008 and 2009 Iran reported ~ 600 dog bites/100,000 individuals [20]. In Africa and Asia, the incidence of exposure to rabies virus through bites or licks by rabid dogs seems to be especially high in travelers with an estimated incidence of 0.4% [21].

Novel PrEP rabies vaccines should induce protective VNA titers after a single immunization. Current rabies vaccines stimulate long-lasting B cell memory; anamnestic responses have been observed for more than a decade following immunization [22]. The same would be expected of a new PrEP vaccine although admittedly such studies would have to be conducted post-licensure. A new PEP rabies vaccine should be more immunogenic and induce neutralizing antibodies faster and at higher levels to reduce the number of vaccine doses preferentially to one and the need for RIG. In either case new rabies vaccines to facilitate their use in developing countries would have to be formulated so that they are stable at ambient temperatures. Production and purification procedures should be simple to eventually allow for their local production in less developed countries. 

Novel vaccine delivery methods such as biodegradable ‘bio-needles’ based on silicon [23] or starch polymers [24] might not only increase thermostability of the vaccine but also reduce cost by avoiding the use of syringes, needles, and vials, and, as was shown with silicon microneedles that were tested with an inactivated influenza virus vaccine in mice, may even increase immunogenicity [25]. The use of controlled release antigen delivery systems is being explored to allow for single immunization regimens, in which the vaccine is release over an extended period of time or at predefined intervals [25,26,27] to circumvent the need for booster immunizations. Such methods still phase technical challenges and have thus far neither reached clinical testing for any vaccine nor have they been explored for rabies vaccines.

## 5. Novel Rabies Vaccine Candidates

A plethora of novel rabies vaccines based mainly on the viral glycoprotein have been tested in animals using mainly PEP regimens. Many of them such as a yeast-derived protein vaccine [28] or peptide vaccines showed lack of complete efficacy and are thus unlikely to progress to clinical trials [29,30]. Others, such as genetically engineered live attenuated rabies viruses, which were shown in animals to be safe and efficacious [31,32] are unlikely to gain public approval. Genetic vaccines based on recombinant viruses or plasmid vectors have yielded promising pre-clinical results, but, due to the delay in onset of antibody responses, caused by the need for transcription and translation of the vaccine antigen, could be developed as PrEP vaccines, but are unsuited for PEP. This chapter focuses on novel vaccines that have yielded sufficiently promising preclinical results to warrant further testing in humans or that have already entered clinical trials.

### 5.1. Vaccines Suited for PEP

#### 5.1.1. Adjuvanted Rabies Vaccines

Adjuvants enhance inflammatory responses that are essential for antigen-driven stimulation of naive B and T cells [33,34]. Some of our current vaccines, such as vaccines to hepatitis B virus or human papilloma viruses contain alum as an adjuvant. Some of the inactivated influenza vaccines contain MF59, an oil-in-water emulsion of squalene oil. Current rabies vaccines do not contain adjuvants, and pre-clinical studies adding alum gave mixed results [35,36]. Clinical trials have been conducted with a rabies vaccine, called PIKA rabies vaccine, containing a second-generation adjuvant based on a Toll-like receptor (TLR)-3 agonist composed of a synthetic dsRNA analogue and a refined form of polyinosinic-polycytidylic acid stabilized with kanamycin and calcium. The rabies vaccine-TLR-3 agonist mixture was tested initially in hamsters and dogs that had been exposed to rabies virus. The PIKA vaccine was more effective in preventing disease compared to the traditional rabies vaccine [37]. The PIKA rabies vaccine was tested in comparison to Rabipur in human volunteers with no prior history of rabies virus exposure or vaccination. The phase I trial showed the vaccine to be well tolerated [38]. In a phase II trial, neutralizing antibody responses to Rabipur, given in a four-dose IM schedule on days 0, 3, 7, and 14, were compared to those elicited by an accelerated regimen of the PIKA rabies vaccine given at two doses on days 0 and 3 and one dose on day 7. A higher percentage of PIKA vaccine recipients achieved VNA titers ≥0.5IU (57.6%) by day 7 after vaccination than those immunized with the conventional rabies vaccine (43.8%) and average VNA titers were also higher by day 7 (0.6 IU in the PIKA vaccine group vs. 0.39 IU in the positive control group). By day 14 responses became similar [39]. These results are very promising but in order to prove the effectiveness of the adjuvant, a study that uses the same regimen with or without adjuvant needs to be conducted.

#### 5.1.2. Protein Vaccines

The rabies virus glycoprotein forms trimers on the virion and most VNAs bind to conformation-dependent epitopes [40], which has made it difficult to develop a correctly folded effective protein-based rabies vaccine. One protein vaccine, produced upon infection of insect cells with a recombinant baculovirus expressing a form of the rabies virus glycoprotein that spontaneously forms nanoparticles after purification, has undergone phase I and II testing in humans and has progressed to a phase III trial. The vaccine is given three times on days 0, 3, and 7. One would assume the vaccine to be safe and immunogenic in humans, but unfortunately none of the trial results have been published by its developer CPL Biologicals (Dholka, Gujarat, India).

#### 5.1.3. Genetically Modified, Inactivated Rabies Virus

Rabies virus can be modified by reverse genetics [41]. Initial modifications aimed to produce a live but highly attenuated vaccine virus that lacked neurovirulence. It was shown that deletions of the genes encoding the phosphoprotein [31] or the matrix protein [32] render rabies virus apathogenic even in immunocompromised mice. Nevertheless, lingering safety concerns may prevent such live attenuated rabies vaccines from gaining public acceptance. In another version, a rabies virus genome was constructed that encoded two copies of the glycoprotein [42]. The virus was shown to grow well, which is a crucial pre-requisite for eventual commercial development. It was more pathogenic than wild-type virus and rapidly induced disease in mice. For immunization studies the virus was inactivated. Mice and dogs injected once with the inactivated glycoprotein-modified vaccine developed higher virus VNA titers compared to mice injected with an equal dose of inactivated wild-type virus. Protective titers of 0.5 IU were reached slightly earlier. In dogs VNA titers were sustained for a one-year observation period. The inactivated glycoprotein-modified vaccine would be expected to meet the stringent safety requirements for vaccines and if its improved immunogenicity can be confirmed in clinical trials is could provide a viable alternative to current vaccine strains provided that additional practical issues, such as genetic stability, scalability, and thermostability don’t pose obstacles to its commercial development.

### 5.2. Vaccines Suited for PrEP

The above-mentioned vaccines could also be used for PrEP. To make their use for mass PrEP in endemic remote areas feasible, they would have to be cost-effective. The only baculovirus-derived protein vaccine that is available for use in humans is a quadrivalent influenza vaccine called Flublok that costs ~ $25–$45, which is, as argued above, well above of what would be affordable in developing counties. One would assume that the adjuvanted PIKA rabies vaccine or an inactivated glycoprotein-modified rabies vaccine would cost about the same as currently used vaccines although costs could be lower if the vaccines would allow for single dose PrEP regimens.

Genetic vaccines, which deliver the gene or a transcript thereof either directly in form of a DNA or an RNA vaccine or upon its packaging into another virus or a bacterium, are conceptionally similar to protein vaccines – both are subunit vaccines that focus the immune response on one antigen of rabies virus. Genetic vaccines have a number of advantages over protein vaccines. They are easy to produce and purify and allow for correct folding of the antigen by talking advantage of the host cell machinery for its production and post-translational modifications. Production is relatively cheap, and DNA and RNA vaccines can be formulated to ensure their thermostability. They are safe and they have the advantage that immune responses are focused on the encoded antigen unlike viral vector vaccines which also elicit responses to antigens of the carrier, which limits their usefulness for homologous booster immunizations. Genetic vaccines carry their own adjuvant. DNA vaccines contain unmethylated CpG-motifs which trigger activation of TLR-9 [43]. This pathway may differ between species, which could in part explain why DNA vaccines that are highly efficacious in mice underperform in humans [44]. Single-stranded RNA engages with TLR-7 and TLR-8 [45,46], while double-stranded RNA, which may contaminate RNA vaccine preparations, also activates TLR-3 [47], retinoic acid-inducible gene (RIG)-I, and melanoma differentiation-associated protein (MDA)-5 pathways [48,49]. Binding of either of these pathogen-associated molecular patterns (PAMPs) to their pathogen recognition receptors (PRRs) starts a signaling cascade that results in proinflammatory cytokine production and an inflammatory response that drives maturation of antigen presenting cells and is thus crucial for the initiation of adoptive immune responses. The main disadvantage of DNA vaccines thus far has been their low immunogenicity in humans [50]. DNA vaccines for rabies virus have far been tested extensively in pre-clinical models in numerous species using both PrEP and PEP protocols [51,52,53,54,55,56,57,58]. In most cases DNA vaccines achieved protective immunity; nevertheless, they have not yet advanced to clinical trials.

#### 5.2.1. RNA Vaccines

An RNA vaccine expressing the rabies virus glycoprotein was tested in a phase I dose escalation study in rabies virus-seronegative human volunteers between 18–40 years of age [59]. The vaccine was either given on days 0, 28, and 56 or on days 0, 7, and 28. Some individuals were boosted 1 year later. The vaccine was given either IM or ID using regular syringes (80 µg, 160 µg, and 320 µg doses for IM and ID, 640 µg for IM), spring-powered ID (80 µg or 160 µg doses) or IM (200 µg or 400 µg doses) injection devises or a CO_2_ powered ID injector (80 µg or 160 µg doses). Most individuals in either of the groups reported adverse local reactions; more than 70% of recipients reported systemic reactions, such as fever, headache, and chills. 5% of vaccine recipients reported grade 3 systemic reactions. One individual developed autoimmune thyroiditis one year after the 3rd dose and one individual in the 640 µg IM dose developed a transient moderate Bell’s palsy 7 days after the second vaccine dose. Overall, the RNA vaccine was more reactogenic than human diploid cell strain rabies vaccines or fowl embryo vaccines, which in the initial trials reported a few cases of mild local side effects but no systemic reactions [60,61]. In the RNA vaccine trial volunteers were tested for rabies-specific VNAs at several timepoints after vaccination. None of the individuals that received the RNA vaccine ID or IM by syringe developed rabies VNA titers ≥ 0.5 IU. Using injection devises 46% of individuals developed protective titers after IM immunization while response rates were slightly higher upon ID immunization (58% and 76% depending on the devise). None of the individuals tested one year later had VNA titers ≥ 0.5 IU and after the boost only 57% increase their titers to these protective levels. Overall this vaccine is not suited for human rabies PrEP; in spite of three doses VNA responses remained below the level needed for protection in a significant percentage of vaccine recipients, immunological memory was poor, and the vaccine caused more frequent and more severe adverse events than currently used rabies vaccines.

#### 5.2.2. Viral Vector Vaccines

Production of viral vectors is more complicated and costly than that of DNA vaccines. The other disadvantage of viral vector vaccines is that as a rule they can only be used once in an individual as VNAs induced against the vector backbone will reduce vaccine uptake and thereby expression and immunogenicity of the vaccine antigen upon its use for a boost [62,63]. Similar to DNA vaccines viral vectors carry PAMPs, which will elicit inflammatory responses needed for initiating adaptive immune responses [64]. Viral vectors have the advantage over DNA vaccines that they enter cells more efficiently upon binding to cell surface receptors, which as a rule renders them more immunogenic. One viral vector vaccine based on a vaccinia virus recombinant expressing the rabies virus glycoprotein termed VR-G is licensed for immunization of wild-live animals [65], another called Purevax Rabies, based on a canarypox virus, is used for vaccination of domestic cats [66]. VR-G is too reactogenic for use in humans as was demonstrated upon inadvertent infection of individuals with contact to the vaccine [67,68]. The rabies canarypox vaccine was tested in humans [69]. It was found to be safe but less immunogenic compared to a traditional inactivated rabies vaccine.

Adenoviruses (Ads) have been vectored to express foreign antigen [70]. The Ad genome can be modified by inserting a gene into the deleted early (E)3 domain which contains genes that are not essential for viral growth. Alternatively, an expression cassette can be placed into the deleted E1 domain. Genes encoded by E1 are crucial for Ad replication and their deletion renders Ad vectors replication-defective. Packaging lines are available that transcomplement the deleted E1 genes thereby allowing for efficient growth of E1-deleted Ad vectors. Ads are common species-specific pathogens that have been isolated from different types of animals ranging from humans to frogs. Most humans become infected with Ads at an early age and develop Ad-specific VNAs which dampen infection with the same virus. Immune responses to vaccines based on common human serotypes of Ad are thus commonly attenuated due to their neutralization [63]. This can be circumvented by the use of vaccine backbones based on Ads isolated from other species such as chimpanzees [63]. Although chimpanzee Ads (AdC) are phylogenetically grouped within human Ads [71] and share many of their characteristics, they neither circulated in humans nor do they show cross-neutralization by antibodies to human serotype Ads [72]. 

A human serotype 5 E3-deleted replication-competent Ad (HAdV-5) vector called ONRAB expressing the glycoprotein of rabies was developed for oral immunization of foxes, raccoons, and skunks. ONRAB although replication-competent in humans cannot replicate in any of these species. Upon confirming that ONRAB given orally was immunogenic in its target species [73], a comparison study with VR-G was conducted along the eastern border between the USA and Canada [74]. Both vaccines were equally effective in immunizing skunks but ONRAB induced significantly higher VNA titers in raccoons. These data suggest that ONRAB may be more suited to combat raccoon rabies, but it needs to be pointed out, that differences in baits and attractants used for ONRAB and VR-G may have confounded the results.

The same vaccine backbone with an E1-deletion expressing the glycoprotein of the Evelyn Rokitniki Abelseth (ERA) strain of rabies was tested in mice and found to afford complete protection after a single immunization [75]; nevertheless, immunogenicity and efficacy were attenuated if mice were initially immunized with an unrelated HAdV-5 vector and therefore carried neutralizing antibodies to the vaccine backbone at the time of their immunization [62]. A replication-defective AdC serotype 68 (AdC68) vector expressing the same antigen was equally effective in mice [76]. This vector after a single low dose given IM provided long-term protection to nonhuman primates against a potentially lethal challenge with street rabies virus [77]. The vector was further modified by replacing the AdC68 E4 open reading frames (orf) 6 and 7 with those of HAd-V5, which increases vector yield on cell lines containing the HAd-V5 E1 during production. Orf6 complexes with a gene product of E1 to facilitate mRNA export and this interaction may be more effective between gene products derived from the same virus. The E1-deleted, E4 modified AdC68rab.gp vector is scheduled for phase I testing in 2020/2021. Studies thus far showed that the vector can be stabilized to permit storage at ambient temperatures [78]. New and improved methods allow for cost-effective production of Ad vectors [79] so that presumable depending on doses required for humans the end product would cost less than $3. This construct if shown to be safe and immunogenic in clinical trials may thus provide a cost-effective alternative to current rabies vaccine for more widespread rabies PrEP.

## 6. Conclusions

The worldwide death toll caused by rabies virus has unfortunately remained remarkably stable over the last two decades and it disproportionally affects children below the age of 15. Vaccination programs for domestic or wildlife animals have successful reduced human rabies in some countries but have failed especially in less developed countries of Africa and Asia. Alternative measures should be contemplated to lessen the impact of rabies on human health such as less expensive and more immunogenic vaccines as discussed in this chapter.

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
