# Peer review of "New Rabies Vaccines for Use in Humans"

_vaccines, 2019, doi:10.3390/vaccines7020054_

Round 1

Reviewer 1 Report

The review article, "New rabies vaccines for use in humans", discusses novel human rabies vaccine candidates that are in late stage pre-clinical testing or have advanced to clinal trials, and their potential to replace currently licensed vaccines. It is appropriately organized and thorough. However, grammatical errors and typos are present throughout; suggest that the author seek review from a colleague or other individual well-versed in English grammar. 

Specific comments/suggestions:

(1) Lines 38, 86, and 89: should read "expensive" (38,86) and "expense" (89), not "expansive" and "expanse", respectively.

(2) The titles of sections 4 and 5 should be improved. For example, the title of section 4, "New rabies vaccines", implies that this section will describe different new vaccine candidates, when in actuality it describes the key attributes/improvements that are needed to improve upon current vaccines; section 5 describes novel rabies vaccine candidates. 

Author Response

I addressed the comments from the reviewers as detailed below. I hope the chapter is now acceptable for publication.

We carefully went through the text and fixed several typos.

From the reviewer: (1) Lines 38, 86, and 89: should read "expensive" (38,86) and "expense" (89), not "expansive" and "expanse", respectively. 

These typos were corrected. 

(2) The titles of sections 4 and 5 should be improved. For example, the title of section 4, "New rabies vaccines", implies that this section will describe different new vaccine candidates, when in actuality it describes the key attributes/improvements that are needed to improve upon current vaccines; section 5 describes novel rabies vaccine candidates. 

Section 4 was retitled and expanded. 

Reviewer 2 Report

"New rabies vaccines for use in humans" by Ertl HCJ provides a comprehensive review of current vaccines and future developments. The review is very well written, clearly presenting the advantages and limitations of different vaccines that are currently in market and under clinical development. 

Just wondering if the author would consider including other concepts in improving vaccine efficacy, like slow release vaccines (to avoid giving multiple boosters), microneedle or nanoparticle based delivery methods, thermostable polymers etc, although sufficient data for rabies vaccine in that format is not available. Overall, the review provides up to date review on different rabies vaccines explained in a very clear format that will be greatly helpful for the audience.

Minor edit : lines 38 and 86 - does the author mean expensive ? 

Author Response

Response 1: The reviewer suggested the addition of some content about novel vaccine delivery systems such as thermostable polymers, slow release formulations, etc. that may circumvent the need for booster immunizations. We added a paragraph to section 4 describing the potential of such new delivery methods

Minor edit:These typos, and others, were corrected.
